# Thermoelectric Properties of Cu_2_Te Nanoparticle Incorporated N-Type Bi_2_Te_2.7_Se_0.3_

**DOI:** 10.3390/ma15062284

**Published:** 2022-03-19

**Authors:** Yong-Jae Jung, Hyun-Sik Kim, Jong Ho Won, Minkyung Kim, Minji Kang, Eun Young Jang, Nguyen Vu Binh, Sang-il Kim, Kyoung-Seok Moon, Jong Wook Roh, Woo Hyun Nam, Sang-Mo Koo, Jong-Min Oh, Jung Young Cho, Weon Ho Shin

**Affiliations:** 1Department of Electronic Materials Engineering, Kwangwoon University, Seoul 01897, Korea; yongjae6945@kw.ac.kr (Y.-J.J.); minkyungkim@kw.ac.kr (M.K.); dmsduddl0212@naver.com (E.Y.J.); smkoo@kw.ac.kr (S.-M.K.); jmoh@kw.ac.kr (J.-M.O.); 2Department of Materials Science and Engineering, University of Seoul, Seoul 02504, Korea; hyunsik.kim@uos.ac.kr (H.-S.K.); sang1.kim@uos.ac.kr (S.-i.K.); 3Department of Chemistry, Kookmin University, Seoul 02707, Korea; ballnet09@kookmin.ac.kr; 4Energy and Environment Division, Korea Institute of Ceramic Engineering and Technology (KICET), Jinju 52851, Korea; rkdalswl5600@naver.com (M.K.); nguyenvubinh20121995@gmail.com (N.V.B.); whnam@kicet.re.kr (W.H.N.); 5Department of Materials Engineering and Convergence Technology, Gyeongsang National University, Jinju 52828, Korea; ksky.moon@gnu.ac.kr; 6School of Nano & Materials Science and Engineering, Kyungpook National University, Sangju 37224, Korea; jw.roh@knu.ac.kr

**Keywords:** thermoelectric, n-type, Cu_2_Te, composite

## Abstract

To develop highly efficient thermoelectric materials, the generation of homogeneous heterostructures in a matrix is considered to mitigate the interdependency of the thermoelectric compartments. In this study, Cu_2_Te nanoparticles were introduced onto Bi_2_Te_2.7_Se_0.3_ n-type materials and their thermoelectric properties were investigated in terms of the amount of Cu_2_Te nanoparticles. A homogeneous dispersion of Cu_2_Te nanoparticles was obtained up to 0.4 wt.% Cu_2_Te, whereas the Cu_2_Te nanoparticles tended to agglomerate with each other at greater than 0.6 wt.% Cu_2_Te. The highest power factor was obtained under the optimal dispersion conditions (0.4 wt.% Cu_2_Te incorporation), which was considered to originate from the potential barrier on the interface between Cu_2_Te and Bi_2_Te_2.7_Se_0.3_. The Cu_2_Te incorporation also reduced the lattice thermal conductivity, and the dimensionless figure of merit *ZT* was increased to 0.75 at 374 K for 0.4 wt.% Cu_2_Te incorporation compared with that of 0.65 at 425 K for pristine Bi_2_Te_2.7_Se_0.3_. This approach could also be an effective means of controlling the temperature dependence of *ZT*, which could be modulated against target applications.

## 1. Introduction

Thermoelectric (TE) technology enables direct solid-state conversion without any moving parts or harmful emissions between heat and electrical energy and shows great potential for applications in waste heat recovery. The conversion efficiency of TE devices depends on the performance of TE materials as represented by the dimensionless figure of merit *ZT* = *S*^2^*σT*/*κ_tot_*, where *σ*, *S*, *T*, and *κ_tot_* are the electrical conductivity, Seebeck coefficient, temperature, and total thermal conductivity, respectively. To realize a high *ZT*, it is desirable to have a high power factor (*S*^2^*σ*) and low thermal conductivity (*κ_tot_*) [1,2]. However, each parameter has a trade-off relation, which makes it difficult to achieve high TE performance for practical applications.

Bi_2_Te_3_-based TE materials are thought to be the only materials that can be considered for cooling or low-temperature energy-harvesting applications. In particular, low-temperature heat below 400 °C accounts for more than 70% of industrial waste heat, which is the case for Bi_2_Te_3_-related material systems [3]. Extensive theoretical and experimental studies such as electronic band structure engineering [4,5], doping [6,7], nano-structuring [8,9], and nanocomposite fabrication [10,11] have been suggested to optimize TE performance. Kim et al. reported a significant reduction in lattice thermal conductivity without any deterioration of the electrical properties by introducing dense dislocation arrays into the simple composition of bulk p-type Bi_0.5_Sb_1.5_Te_3_ and obtained the highest *ZT* value of ~1.9 near room temperature [12]. Although significant progress in thermal conductivity reduction has been attained [12,13,14], it remains necessary to boost the electrical properties of TE materials to achieve efficient power generation and cooling devices. The carrier filtering mechanism alters the Seebeck coefficient by introducing interfaces between the TE matrix and secondary nanophases, and the band bending on the interfaces induces the filtering of low-energy carriers [15,16,17,18,19,20]. The secondary nanophases also strengthen the phonon scattering to reduce thermal conductivity, which could significantly enhance *ZT*. The effect of the nano-phase in TE materials has been reported on PbTe-based TE materials, where the SrTe nano-precipitate enhances *ZT* up to ~2.5 at high temperatures [18]. For Bi_2_Te_3_-based materials, the addition of TE to p-type Bi_0.5_Sb_1.5_Te_3_ thin films prepared by a pulsed laser deposition technique significantly enhances the Seebeck coefficient [19]. However, the nano-precipitate and vacuum deposition approaches require delicate processes that cannot be applied practically. In this study, we introduced Cu_2_Te nanoparticles (NPs) into an n-type Bi_2_Te_2.7_Se_0.3_ (BTS) matrix to enhance TE performance. Several papers [21,22,23] have reported the existence of Cu_2_Te phase when excess Cu is incorporated on Bi-Te-based TE materials. However, they do not discuss the direct effect of Cu_2_Te incorporation. Cu_2_Te NPs, which were synthesized by the organic-free chemical method, could be considered as efficient additives that would provide benefits such as modulating the carrier concentration and enhancing phonon scattering. In addition, the incorporation of Cu_2_Te NPs only requires a simple process that can easily be employed to alter TE properties according to the target applications.

## 2. Materials and Methods

The BTS matrix was synthesized using a conventional melting-quenching process. Raw elements with stoichiometric ratios (Bi, 99.999%, 5 N Plus; Te, 99.999%, 5 N Plus; Se, 99.999%, 5 N Plus) were heated at 1000 °C for 6 h under a vacuum in fused silica tubes, and the melts were quenched in a water vessel and finally ground into fine powders using ball milling. 

The Cu_2_Te NPs were synthesized using the chemical reduction method. Under an Ar atmosphere, 1 g Te was dissolved in 0.3 M NaBH_4_ aqueous solution. Further, 0.2 M CuCl_2_·H_2_O aqueous solution was slowly poured into the above Te solution and stirred for 30 min. After remaining for an additional 30 min, the precipitates were centrifuged and washed with ethanol several times. For the Cu_2_Te NP dispersion on the BTS matrix, the obtained Cu_2_Te NPs were dispersed in ethanol and mixed with BTS by wet grinding. The mixture was dried in a vacuum oven, and fine powders were collected. The obtained powders were sintered by spark plasma sintering at 773 K for 3 min under a pressure of 60 MPa. 

The consolidated samples were analysed by X-ray diffraction (XRD, Cu Kα, 1.5406 Å, New D8 Advance, Bruker, Billerica, MA, USA), scanning electron microscopy (SEM, JSM-7600F, JEOL, Tokyo, Japan), electron probe microanalysis (EPMA, 30 kV, JXA-8530F PLUS, JEOL, Tokyo, Japan), and transmission electron microscopy (TEM, 200 kV, Tecnai G2-20, FEI, Hillsboro, Oregon, USA). The TE properties were measured using a ZEM-3 (ULVAC-RIKO, Methuen, MA, USA) for the electrical parts and the laser-flash method (LFA, DLF 1300, TA, New Castle, DE, USA) for the thermal parts. Hall measurements were conducted to obtain the carrier concentration (HT-Hall, ResiTest 8300, Toyo Corporation, Tokyo, Japan).

## 3. Results and Discussions

Figure 1a shows the XRD pattern of as-prepared Cu_2_Te NPs obtained by the chemical reduction method. Our Cu_2_Te NPs has a hexagonal structure with the space group of P3m1 (JCPDS # 49-1441). The XRD peaks are not strong due to the small size of Cu_2_Te NPs. The size of as-prepared Cu_2_Te is 20–50 nm from the SEM and TEM images (Figure 1b,c), and the high resolution TEM image shows the lattice distance of 0.36 nm, which corresponds to (006) plane of Cu_2_Te phase. Figure 1e presents the powder XRD patterns of the BTS-*x* wt.% Cu_2_Te NP (*x* = 0, 0.2, 0.4, 0.6, and 0.8) composite materials. All samples show typical BTS patterns (rhombohedral structure, space group of R-3m, JCPDS # 50-0594) without any additional peaks. The lattice parameters of the BTS-*x* wt.% Cu_2_Te NP (*x* = 0, 0.2, 0.4, 0.6, and 0.8) are summarized in Table 1, and there are no significant changes in lattice parameters by Cu_2_Te NPs incorporation. The small size and low content of Cu_2_Te NPs in the composite powder were not detectable by XRD (Figure 1e). We analysed the Cu_2_Te nanophases on BTS-*x* wt.% Cu_2_Te NP (*x* = 0.2, 0.4, and 0.6) bulk samples by EPMA (Figure 1f–h), where the light parts of the Cu mapping were matched with the Cu_2_Te nanophases. Cu_2_Te NPs can be easily observed on the parent bulk materials, and as the amount of Cu_2_Te NPs increases, a homogeneous distribution of Cu_2_Te NPs can be found. However, for the BTS-0.6 wt.% Cu_2_Te NPs sample, some Cu_2_Te NPs exhibit agglomeration, which means that a large amount (more than 0.6 wt.%) of Cu_2_Te NPs on BTS cannot be dispersed uniformly. 

Figure 2a presents the temperature dependence of the electrical conductivity (*σ*) for BTS-*x* wt.% Cu_2_Te NPs (*x* = 0, 0.2, 0.4, 0.6, and 0.8). The electrical conductivity gradually decreases over the entire temperature range studied here as the Cu_2_Te NP incorporation increases. The electrical conductivities of BTS-*x* wt.% Cu_2_Te NPs (*x* = 0, 0.2, 0.4, and 0.6) decrease with increasing temperature, indicating semi-metallic or metallic conduction behaviour. However, the sample with a high amount of Cu_2_Te NPs (0.8 wt.%) incorporated shows semiconducting behaviour at a high temperature due to the carrier concentration decrease shown in Figure 2b. The carrier concentration obtained from the Hall measurements decreases as the amount of Cu_2_Te NPs increases. The Cu intercalation in the van der Waals gap between Te–Te shows donor-like behaviour, while our Cu_2_Te incorporation shows acceptor-like behaviour. The temperature dependence of the Seebeck coefficient (*S*) for the BTS-*x* wt.% Cu_2_Te NP (*x* = 0, 0.2, 0.4, 0.6, and 0.8) samples is displayed in Figure 2c. All samples exhibit negative *S* values, indicating that electrons constitute the majority of the charge carriers, which is consistent with the signs of the respective Hall measurements. The temperature at which the *S* peaks (*T_max_*) are shifted to lower temperatures with increasing Cu_2_Te incorporation, and the absolute value of S maximum (*S_max_*) increases as the amount of incorporated Cu_2_Te increases. Typically, BTS show anisotropic nature of TE properties due to Te vacancies and antisite defects [24]. It is noted that the reproducibility (collecting the results for more than 5 different batches) of electrical properties is greatly enhanced through Cu_2_Te introduction. 

Usually, the *T_max_* shift to lower temperatures originates from increased bipolar conduction, but the corresponding *S_max_* should also be decreased. Improvement of the bipolar conduction cannot explain the observed increase in *S_max_* with increasing Cu_2_Te. From Pisarenko’s relation [1,25], one can estimate the density of states (DOS) effective mass with the assumption of a single parabolic band model using the following Equation (1):(1)$$S=\frac{8\pi^{2}k_{B}^{2}}{3eh^{2}}{\left(\frac{\pi }{3n}\right)}^{2/3}m_{d}^{*}T$$
where *h* is the Planck constant, *k_B_* is the Boltzmann constant, $m_{d}^{*}$ is the DOS effective mass, *e* is the electronic charge, and *n* is the carrier concentration. As shown in Figure 2e, the pristine BTS has an $m_{d}^{*}$ value of 0.97 *m_e_* (*m_e_* is the electron rest mass), whereas Cu_2_Te NP incorporation increases the $m_{d}^{*}$ value of 1.16 *m_e_* in the BTS-0.4 wt.% Cu_2_Te NP sample at room temperature. Therefore, the increase in $m_{d}^{*}$ with Cu_2_Te incorporation can enhance *S_max_*, even when the bipolar conduction becomes strong. The *n* reduction observed with the incorporation of Cu_2_Te is also responsible for the increase in *S_max_*. The $m_{d}^{*}$ change could be attributed to the engineered band structure interface between the BTS matrix and Cu_2_Te NPs. In this context, we can suggest a band diagram that describes the interfacial band bending between the BTS and Cu_2_Te NPs (Figure 2f). The electron energy barrier, i.e., the hetero-interface of the conduction bands between the BTS and Cu_2_Te NPs, filters the low-energy carriers. The electron affinity and band gap of BTS and Cu_2_Te were obtained from the literature [26,27,28,29]. A previous theoretical study of PbTe TE materials showed that ~1.5 nm nanoinclusion significantly enhanced the *S* value [15]. The electrostatic potential only affected the interface between the nanoinclusion and the matrix, and larger nanoinclusions were less effective than smaller nanoinclusions. Our synthesized Cu_2_Te NPs had sizes of ~50 nm, so we expected that the smaller Cu_2_Te NPs could have a greater effect. The BTS-0.4 wt.% Cu_2_Te NP sample has the highest calculated power factor (*S*^2^*σ*), which is ~15% higher at room temperature than that of the pristine BTS sample.

The temperature dependence of the total thermal conductivity (*κ**_tot_*) of BTS-*x* wt.% Cu_2_Te NPs (*x* = 0, 0.2, 0.4, 0.6, and 0.8) is shown in Figure 3a. All samples display very low *κ**_tot_* values over the entire temperature range studied here and also exhibit upturns. The occurrence temperatures of the upturns in the thermal conductivity data move to lower temperatures with increasing Cu_2_Te incorporation amounts, which is consistent with the behaviour of *S* and *σ*. This increase in thermal conductivity is due to the bipolar diffusion of the carriers and is still present in the BTS sample at elevated temperatures, as shown in Figure 3a. This increase in *κ**_tot_* with temperature is attributable to the thermal energy transported by electron–hole pairs, which is equal to the energy of the gap between the hot and cold sides of the sample. This bipolar diffusion phenomenon results in an increase in the heat transfer beyond what is expected from the normal carrier contribution (*κ**_ele_*) to *κ* defined by the Wiedemann-Franz law [30], *κ**_ele_* = *LσT*, where *L* is the Lorenz number. The Lorenz number is estimated by assuming a single parabolic band and acoustic phonon scattering using the following Equations (2)–(4):(2)$$S=\pm \frac{k_{B}}{e}\left[\frac{\left(r+5/2\right)F_{r+3/2}\left(\xi \right)}{\left(r+3/2\right)F_{r+1/2}\left(\xi \right)}-\xi \right]$$
(3)$$F_{n}\left(\xi \right)=\int_{0}^{\infty }\frac{x^{n}}{1+e^{\left(x-\xi \right)}}dx$$
(4)$$L={\left(\frac{k_{B}}{e}\right)}^{2}\left[\frac{\left(r+7/2\right)F_{r+5/2}\left(\xi \right)}{\left(r+3/2\right)F_{r+1/2}\left(\xi \right)}-{\left(\frac{\left(r+5/2\right)F_{r+3/2}\left(\xi \right)}{\left(r+3/2\right)F_{r+1/2}\left(\xi \right)}\right)}^{2}\right]$$

Here, *ξ*, *F_n_*(*ξ*), and *r* are the reduced Fermi energy ((*E_v_* − *E_F_*)/*k_B_T*), Fermi integral of order *n*, and scattering parameter, respectively. We set *r* = 0–1/2 for acoustic phonon scattering. The temperature dependence of the lattice thermal conductivity (*κ_lat_*) can be estimated by deducting the bipolar conduction portion (*κ**_bp_*) and electronic portion (*κ**_ele_*) of thermal transport from the total thermal conductivity; *κ_lat_* = *κ_tot_* − *κ_bp_* − *κ_ele_*. *κ_bp_* was calculated using the below Equation (5).
(5)$$\kappa_{bp}=\left(\sum_{i}S_{i}{}^{2}\sigma_{i}-S^{2}\sigma \right)T.$$

*S_i_*, *σ_i_*, *S*, and *σ* are the Seebeck coefficient and electrical conductivity of an individual band, total Seebeck coefficient, and electrical conductivity from both the conduction and valence bands, respectively. *S* and *σ* are in turn defined as follows Equations (6) and (7):(6)$$\sigma =\sum_{i}\sigma_{i}$$
(7)$$S=\frac{\sum_{i}S_{i}\sigma_{i}}{\sum_{i}\sigma_{i}}$$

Using the two-band (TB) model (an extension of the single parabolic band model), which includes one valence band and one conduction band, band parameters such as the density-of-states effective mass ($m_{d}^{*}$) and non-degenerate mobility (*μ*_0_) of each band were obtained. Specifically, the $m_{d}^{*}$ and *μ*_0_ of individual band were estimated by fitting the TB model to Hall carrier concentration (*n_H_*)-dependent *S* and *n_H_*-dependent *σ* measurements, respectively [31]. The band gap between the valence and conduction bands required in the TB model was adopted from the literature [32]. Once $m_{d,i}^{*}$ and *μ*_0,*i*_ (*i* = valence and conduction bands) were estimated, corresponding *S_i_* and *σ_i_* (*i* = valence and conduction bands) were calculated. Furthermore, theoretical total *S* and *σ* (which agree well with experimental *S* and *σ*) were computed with *S_i_* and *σ_i_* according to Equations (6) and (7). Finally, the calculated *S_i_*, *σ_i_*, *S*, and *σ* were substituted back into Equation (5) to estimate *κ_bp_*.

The incorporation of Cu_2_Te NPs encourages bipolar conduction and, consequently, *κ_lat_* (shown in Figure 3b), after Cu_2_Te NP incorporation is reduced over the entire temperature range due to enhanced phonon scattering. The reduction in *κ_lat_* occurs up to 0.4 wt.% Cu_2_Te NPs incorporation, whereas more Cu_2_Te NPs produce a negative phonon scattering effect by agglomeration of Cu_2_Te NPs. Generally, it is acceptable that the introduction of secondary phases decreases *κ_lat_* by phonon scattering. However, the size of the secondary phase is an important factor for enhancing phonon scattering. Particles with sizes larger than hundreds of nanometres have a limited effect on *κ_lat_* [33]. Our *κ_lat_* results after Cu_2_Te agglomeration (more than 0.6 wt.% Cu_2_Te NPs) could be described similarly. The size of the Cu_2_Te NPs was ~50 nm, and if several Cu_2_Te NPs were agglomerated, the size was easily greater than a hundred nanometres. In addition, the previous report also shows that reduced particle size with the same volume fraction could decrease thermal conductivity more [34]. Therefore, a homogeneous dispersion of small Cu_2_Te NPs is necessary for encouraging phonon scattering and enhancing the electrical properties. 

Collecting the above effects, *ZT* is shown as a function of temperature for BTS-*x* wt.% Cu_2_Te NPs (*x* = 0, 0.2, 0.4, 0.6, and 0.8) in Figure 3c. The highest *ZT* of 0.75 is observed at 374 K for the BTS-0.4 wt.% Cu_2_Te NPs sample, which is 15% higher than that of the pristine sample. It is noteworthy that the maximum *ZT* temperature for each sample is modulated by the introduction of Cu_2_Te NPs, decreasing as the amount of Cu_2_Te NPs increases. The simple incorporation of Cu_2_Te can easily modulate the temperature dependence of n-type TE materials, and we suggest that the TE properties can be precisely controlled according to the target application, such as cooling or low-temperature energy harvesting. The average *ZT* value (*ZT_avg_*) in the temperature range studied here is shown in Figure 3d. *ZT_avg_* reaches 0.70 for the BTS-0.4 wt.% Cu_2_Te NPs sample, representing 19% enhancement compared to the pristine BTS matrix. 

## 4. Conclusions

In conclusion, we investigated the TE properties of n-type BTS by incorporating Cu_2_Te NPs. The incorporation of Cu_2_Te NPs encourages the Seebeck coefficient and power factor, and we thought that the band bending between the interface of the BTS and Cu_2_Te NPs could filter the low-energy electron carrier. The Cu_2_Te NPs on the BTS matrix also affect *κ_lat_* by encouraging phonon scattering. Together with modulating the electrical and thermal properties, the maximum *ZT* value reaches 0.75 at 374 K for the BTS-0.4 wt.% Cu_2_Te NPs sample, 15% higher than that of pristine BTS (0.65 at 425 K). In addition, the temperature dependence of *ZT* can be controlled by the Cu_2_Te NP dispersion just before the sintering process, which is a great advantage for target applications. Further advances could be expected by developing the synthesis of Cu_2_Te NPs with sizes of several nanometres.

## Figures and Tables

**Figure 1 materials-15-02284-f001:**
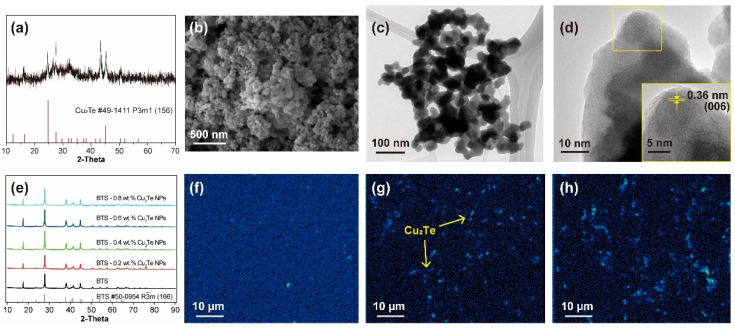
(**a**) XRD pattern, (**b**) SEM image, (**c**) TEM image, and (**d**) high resolution TEM image of Cu_2_Te NPs, (**e**) XRD patterns of BTS-*x* wt.% Cu_2_Te NPs (*x* = 0, 0.2, 0.4, 0.6, and 0.8), Cu K mapping images of EPMA for (**f**) BTS-0.2 wt.% Cu_2_Te NPs, (**g**) BTS-0.4 wt.% Cu_2_Te NPs, and (**h**) BTS-0.6 wt.% Cu_2_Te NPs, where the light dots correspond to elemental Cu.

**Figure 2 materials-15-02284-f002:**
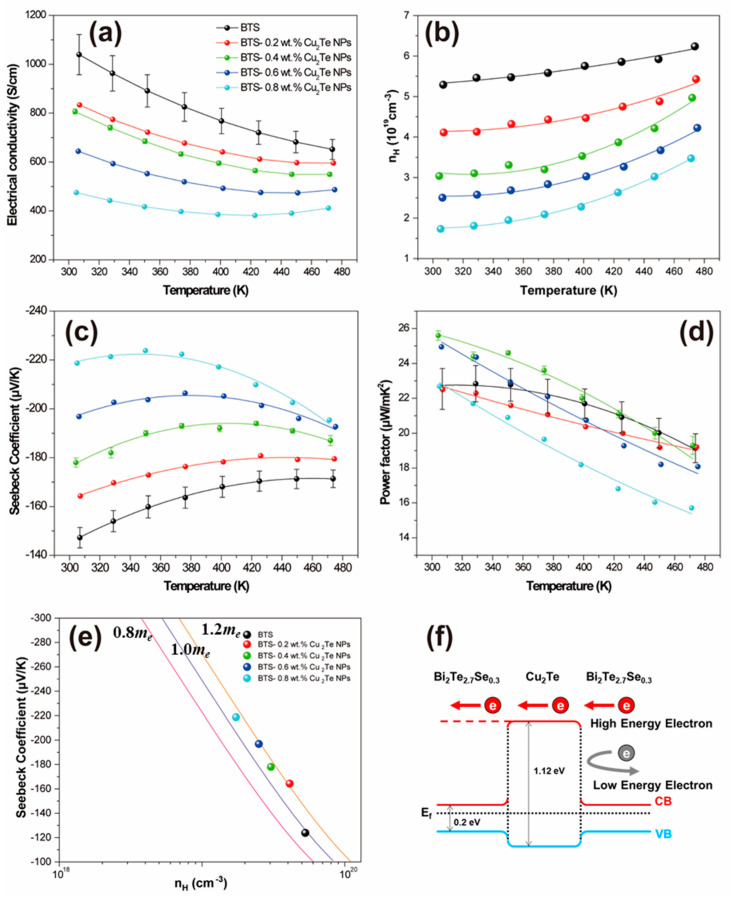
Electrical properties of BTS-*x* wt.% Cu_2_Te NPs (*x* = 0, 0.2, 0.4, 0.6, and 0.8). (**a**) Electrical conductivities, (**b**) electron carrier concentrations obtained by Hall measurement, (**c**) Seebeck coefficients, (**d**) power factors, (**e**) Pisarenko’s plot, and (**f**) band diagram of BTS and Cu_2_Te NPs interfaces.

**Figure 3 materials-15-02284-f003:**
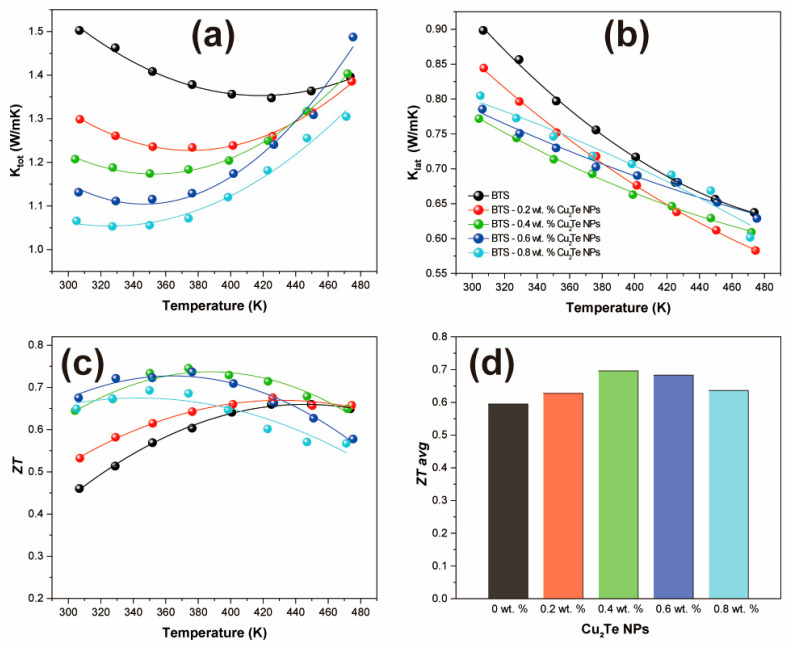
(**a**) Total thermal conductivities, (**b**) lattice thermal conductivities, (**c**) *ZT* values, and (**d**) average *ZT* values between room temperature and 470 K of BTS-*x* wt.% Cu_2_Te NPs (*x* = 0, 0.2, 0.4, 0.6, and 0.8).

**Table 1 materials-15-02284-t001:** The lattice parameters of BTS-*x* wt.% Cu_2_Te NPs (*x* = 0, 0.2, 0.4, 0.6, and 0.8).

	Lattice Constant for a (Å)	Lattice Constant for c (Å)
BTS	4.3619	30.3632
BTS-0.2 wt.% Cu_2_Te NPs	4.3614	30.3604
BTS-0.4 wt.% Cu_2_Te NPs	4.3615	30.3641
BTS-0.6 wt.% Cu_2_Te NPs	4.3610	30.3606
BTS-0.8 wt.% Cu_2_Te NPs	4.3614	30.3650

## Data Availability

The data presented in this study are available upon request from the corresponding author.

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
