# Peer review of "Thermoelectric Properties of Cu2Te Nanoparticle Incorporated N-Type Bi2Te2.7Se0.3"

_materials, 2022, doi:10.3390/ma15062284_

Round 1

Reviewer 1 Report

The authors studied the dispersion effect of Cu2Te nanoparticles (NPs) on the thermoelectric properties of Cu2Te NPs decorated Bi2Te2.7Se0.3. They showed that with an optimal dispersion and loading of Cu2Te NPs, the dimensionless figure of merit ZT could be improved compared to the pristine Bi2Te2.7Se0.3. This is generally a solid work but a few issues listed below need to be addressed before it can be accepted for publication:

  1. The size analysis of the synthesized Cu2Te NPs is very rough. TEM is needed for better characterization of the size distribution and HR-TEM is recommended for the phase analysis since the XRD pattern shown in Figure 1a is really noisy.
  2. In Figure F, is the number 1.1.2 eV a typo? How did the authors measure or calculate the bind gap of their synthesized materials?

Author Response

We appreciate for the reviewer's positive response. Following the reviewer’s comment, we have progressed additional experiments and analysis and revised the manuscript.

Reviewer 2 Report

The manuscript " Thermoelectric properties of Cu2Te Nanoparticle Incorporated N-type Bi2Te2.7Se0.3" by Y.-J. Jung et al is suitable for publication in the journal “Materials” with minor corrections.

The manuscript presents the investigations of the thermoelectric properties of alloys of composition Bi2Te2.7Se0.3 with the addition of different amounts (0 - 0.8 wt.%) of nanoparticles of the phase Cu2Te. The motivation, presentation of the state of research, description of the sample preparation and characterization as well as the evaluation of the measurement results are presented in a compact and comprehensible way. The discussion and interpretation of the measurement results are comprehensible. In the conclusion, it is pointed out that the thermoelectric properties can be specifically adjusted with the proposed preparation method.

The possibility to adjust the thermoelectric properties is the special value of these investigations and should therefore be clearly stated in the manuscript. More precise information on the reproducibility and possible variations of the measurement results in samples with formally the same composition and preparation would underline the importance of the investigation results.

Specifically, the following changes should be made:

Line 101: The phase Cu2Te should not only be characterized by the JCPDS number. The space group and the lattice parameter reported in literature should be given. Since several modifications of Cu2Te exist, it is important to define exactly which variant has been detected.

Line 103: Bi2Te2.7Se0.3 should also not only be characterized by the JCPDS number. The space group and lattice parameter should be given.

The determination and discussion of the lattice parameters of the Bi2Te2.7Se0.3 phase is desirable. At least the behaviour of the lattice parameters as a function of the concentration of the Cu2Te fraction should be discussed qualitatively. The lattice parameter should also be compared with literature data.

Figure 1a) The red lines in the inset are hardly visible. A comparison with the experimental diffraction pattern is therefore hardly possible. A different form of presentation should be chosen here. Presumably, the diffraction image shows some diffraction lines of the phase Cu2Te and confirms the existence of this phase. The diffraction image shows - presumably - even further diffraction lines. Therefore, it should be pointed out in the text that the existence of secondary phases cannot be excluded.

Figure 1a, and b) indicate the wavelength of the X-rays.

Figure caption 1 c)-e): the excitation voltage of the electron microscope and the Cu line (Cu K or Cu L) should be given.

Line 124: “The carrier concentration obtained from the Hall measurements decreases as the amount of Cu2Te NPs increases”.

This behaviour is very remarkable and should be discussed. An increase in electron concentration with increasing Cu content is expected and is observed in other studies (e.g. in [24] Figure 4 d). This difference should be mentioned and commented on if possible.

Line 131: “….. S maximum (Smax) increases as the amount of incorporated Cu2Te increases Smax increases”

This description is misleading. Does only the value of Smax change or does the difference between minimum and maximum value in the investigated interval change? Please clarify?

Line 133: “It is noted that the reproducibility of electrical properties is greatly enhanced through Cu2Te introduction”

Does this also apply to the samples in this investigation? A comment on the reproducibility of the presented results should be given here.

Lines 210 - 216: This section is worded in an ambiguous way and should be revised. Here it is important to distinguish between experimental values (Sexp sexp,) calculated values((Scalc scalc,) and values of the reference material (Sref , sref) . Overall, an interesting approach that is unfortunately hard to understand.

Line 245: “The band bending between the interface of the BTS and Cu2Te NPs filters the low-energy electron carrier, enhancing the Seebeck coefficient and power factor”.

This formulation should be made weaker. There are reasonable arguments that band bending at the interface play a role but this is only a model.

Author Response

We appreciate for the reviewer's valuable comments and suggestions to clarify main idea in this work. Following the reviewer’s comment, we have progressed additional experiments and analysis and revised the manuscript.
